# The First Genetic Linkage Map of Winged Bean [*Psophocarpus tetragonolobus* (L.) DC.] and QTL Mapping for Flower-, Pod-, and Seed-Related Traits

**DOI:** 10.3390/plants11040500

**Published:** 2022-02-12

**Authors:** Sompong Chankaew, Sasiprapa Sriwichai, Teppratan Rakvong, Tidarat Monkham, Jirawat Sanitchon, Sithichoke Tangphatsornruang, Wasitthee Kongkachana, Chutima Sonthirod, Wirulda Pootakham, Kitiya Amkul, Anochar Kaewwongwal, Kularb Laosatit, Prakit Somta

**Affiliations:** 1Department of Agronomy, Faculty of Agriculture, Khon Kaen University, Khon Kaen 40002, Thailand; somchan@kku.ac.th (S.C.); sasiprapa.sr@kkumail.com (S.S.); teppratan_r@kkumail.com (T.R.); tidamo@kku.ac.th (T.M.); jirawat@kku.ac.th (J.S.); 2National Omics Center (NOC), National Science and Technology Development Agency, 111 Thailand Science Park, Khlong Nueng, Khlong Luang, Pathum Thani 12120, Thailand; sithichoke.tan@nstda.or.th (S.T.); wasitthee.kon@nstda.or.th (W.K.); chutima.son@nstda.or.th (C.S.); wirulda.poo@nstda.or.th (W.P.); 3Department of Agronomy, Faculty of Agriculture at Kamphaeng Saen, Kasetsart University, Nakhon Pathom 73140, Thailand; fagrkia@ku.ac.th (K.A.); anochar.au@gmail.com (A.K.); fagrkal@ku.ac.th (K.L.)

**Keywords:** RAD-seq, SNPs, pod length, seed color, winged bean, anthocyanin

## Abstract

Winged bean [*Psophocarpus tetragonolobus* (L.) DC.] (2n = 2× = 18) is a tropical legume crop with multipurpose usages. Recently, the winged bean has regained attention from scientists as a food protein source. Currently, there is no breeding program for winged bean cultivars. All winged bean cultivars are landraces or selections from landraces. Molecular markers and genetic linkage maps are pre-requisites for molecular plant breeding. The aim of this study was to develop a high-density linkage map and identify quantitative trait loci (QTLs) for pod and seed-related traits of the winged bean. An F_2_ population of 86 plants was developed from a cross between winged bean accessions W054 and TPT9 showing contrasting pod length, and pod, flower and seed colors. A genetic linkage map of 1384 single nucleotide polymorphism (SNP) markers generated from restriction site-associated DNA sequencing was constructed. The map resolved nine haploid chromosomes of the winged bean and spanned the cumulative length of 4552.8 cM with the number of SNPs per linkage ranging from 36 to 218 with an average of 153.78. QTL analysis in the F_2_ population revealed 31 QTLs controlling pod length, pod color, pod anthocyanin content, flower color, and seed color. The number of QTLs per trait varied between 1 (seed length) to 7 (banner color). Interestingly, the major QTLs for pod color, anthocyanin content, and calyx color, and for seed color and flower wing color were located at the same position. The high-density linkage map QTLs reported in this study will be useful for molecular breeding of winged beans.

## 1. Introduction

The winged bean [*Psophocarpus tetragonolobus* (L.) DC.] (2n = 2× = 18) is a tropical legume crop belonging to the family Fabaceae and subfamily Papilionoideae [1]. The crop originated in South-East Asia or perhaps Papua New Guinea. It is widely grown in hot humid equatorial countries throughout Southeast Asia and East Africa [2], and is an important leguminous vegetable crop in Thailand, Burma, Laos, Malaysia, Vietnam, Indonesia, Bangladesh, Sri Lanka, Ghana, and Nigeria [2]. Although the young pod of the winged bean is the most popularly consumed part, the rest of the plant parts are also edible when appropriately prepared [3]. Winged bean seeds contain high protein (30–37%) and oil (15–18%) contents [4,5]. In addition, the winged bean tuber contains roughly 20% protein and 25–30% carbohydrates [6]. As a viable candidate for diversifying diets, roasted or boiled tuberous roots are capable of improving the nutritional security of the people in tropical regions [6]. The crop has received scientific praise for its nutritional content, as comprehensively described in ‘The winged bean: high-protein crop for the humid tropics’ from the National Academy of Science in 1981 [7]. The winged bean is one of the most important legume crops due to its high protein value and multipurpose usage. As a vegetable legume crop, the winged bean holds a unique position in the world of food and agriculture [5]. It has been introduced to more than 80 countries worldwide [8].

The availability and assessment of genetic variability is a prerequisite for crop improvement. The centers of origin and diversity of the winged bean are not yet known. In addition, the ancestors of the winged bean are still not known and may be extinct [9]. However, winged beans from Papua New Guinea and Indonesia have been reported to be diverse, while increasing numbers of winged bean landraces have recently been discovered in Thailand [2,10]. Studies of genetic diversity have demonstrated that the winged bean has moderate genetic variation [5,8,10,11,12].

All winged bean cultivars planted today are indigenous or local cultivars. The demand for winged bean production is increasing, due to its security as a staple food source and multipurpose usage [13]. Therefore, cultivars possessing higher yields with improved traits are desirable. However, knowledge on the genetics and breeding of the winged bean is very limited. Most previous studies in this crop focused on production systems and variation in morphological traits [6,14,15]. 

Very limited information on the genetics and genomics of the winged bean has been reported. The genome size of winged beans is large (2n = 2× = 18; 1.22 Gbp/C) [11], and until now a genetic linkage map of the winged bean is not available. All previous studies of molecular genetics and genomics of the winged bean have been related to genetic diversity using first and/or second-generation DNA markers (RAPD and ISSR markers [5], ISSR [8], and SSR [10]). As a result, there exists no study on the molecular breeding of the winged bean, due to a lack of genomic tools.

A genetic linkage map is an indispensable tool for molecular breeding of crops as it is necessary identify the quantitative trait loci (QTLs) or genes controlling traits in breeding programs which will be ultimately useful for marker-assisted selection. To construct a genetic linkage map, DNA markers must be developed. Although some SSR and SNP markers have been developed for the winged been [11,12], not many of them have been validated for use in molecular analysis. In addition, the analysis of SSR and SNP markers in winged bean is expensive, laborious, and difficult because of its large genome size. Recently, next-generation sequencing (NGS) technologies have accelerated whole-genome sequencing and genotyping [16]. Whole genomes of numerous legume species, including mungbean (*Vigna radiata*) [17], cowpea (*V. unguiculata* [L.] Walp.) [18], beach pea (*V. marina*) [19], adzuki bean (*V. angularis*) [20], pea (*Pisum sativum* L.) [21], pigeonpea (*Cajanus cajan*) [22], chickpea (*Cicer arietinum*) [23], peanut (*Arachis ipaensis*) [24], and soybean (*Glycine max*) [25] have been sequenced. Genotyping using high-throughput NGS technologies reduces the complexity of genome analysis and allows the development of numerous SNP markers for any species of interest [26]. Restriction-site associated DNA sequencing (RAD-seq), including double digested RAD-seq (ddRAD-seq), is a preferable NGS-based technology for high-throughput genotyping [16,27]. The RAD-seq method, first described by Baird et al. [28], can be employed without a reference genome, and a reference-guided RAD-seq approach can improve the accuracy of SNP detection.

In this study, we aimed to develop the first genetic linkage map and identify QTL controlling flower-, pod- and seed-related traits in the winged bean. The map was constructed using SNP markers generated by the RAD-seq method. The genetic linkage map and the QTL information generated from this study will facilitate the molecular breeding of winged beans in future studies.

## 2. Results

### 2.1. Variation in Pod Anthocyanin Content, Pod Length, Seed Length, and Seed Width 

The parents, TPT9 and W054, of the F_2_ mapping population used in this study were clearly different in pod anthocyanin content (pod color) and pod length, but only slightly different in seed size (length and width). Pod anthocyanin content of W054 was 17 times higher than that of TPT9. The value of the content was 40.93 and 2.40 mg/100 g for the former and the latter, respectively (Figure 1A). The values of the pod anthocyanin contents in the F_2_ population ranged from 1.91 to 34.28 mg/100 g, with an average of 9.08 mg/100 g. Pod length of the W054 and TPT9 was 24.29 and 15.06 cm, respectively. The pod length in the F_2_ population varied from 14.25 to 26.50 cm with an average of 19.25 cm (Figure 1B). TPT9 and W054 produced seed length of 8.21 and 9.22 mm, respectively (Figure 1C). Seed length in the F_2_ population varied between 7.67 and 10.70 mm, with an average of 9.43 mm. Seed width of TPT9 was 7.89 mm, while that of W054 was 7.13 mm. However, the seed width in the F_2_ population ranged from 5.82 to 9.10 mm, with an average of 7.76 mm (Figure 1D). Variation of pod anthocyanin content, pod length, seed length, and seed width in the F_2_ population was continuous and normally distributed (Figure 1) (Durbin-Watson statistic test with *p*-values of 0.7053, 0.2435, 0.9836, and 0.1926, respectively—data not shown), suggesting that they are quantitative traits and possibly controlled by polygenes. 

### 2.2. Variation in Organ Coloration

TPT9 and W054 were contrastive in pod, seed, and flower (calyx, wing, and banner) color (Figure 2). All of these traits except seed coat color showed high variation in the F_2_ population. For example, pod colors of TPT9 and W054 were blue and purple, respectively, while pod colors in the F_2_ population were green, green with purple spots, green with a purple spot at center, purple wing, purple center, light purple, purple, and dark purple (Figure 2A). Interestingly, wing and banner colors showed transgressive segregation; some colors not existing in both parents. Classes and patterns of segregation of wing color and banner color in the F_2_ population were nearly the same (Figure 2D,E), suggesting that colorations of these two organs were controlled by the same genes or tightly linked genes. Nonetheless, these results suggest that coloration of pod, seed, and flower (calyx, wing, and banner) color is controlled by two or more genes in which some of the genes have pleotropic effect or are linked. 

Results of Pearson’s correlation coefficient analysis of the nine flower-, pod-, and seed-related traits in the F_2_ population are presented in Table 1. The anthocyanin content was significantly correlated with pod color (0.748) and calyx color (0.706). Pod color and flower color were also significantly correlated, as were pod color and calyx color (0.812), calyx color and wing color (0.543), calyx color and banner color (0.523). The highest correlation was found between the wing color and banner color (0.978) (Table 1). Moreover, seed color was also significantly correlated with pod color (0.547) and calyx color (0.595) (Table 1). The significant correlation among flower, pod, and seed colors indicates, again, that colorations of these organs are controlled by the same or closely linked gene(s). 

### 2.3. Construction of the Winged Bean Genetic Linkage Map

#### 2.3.1. RAD-Seq Library Construction, Sequencing, and SNP Calling 

In this study, a RAD-seq library was constructed using the MGIEasy RAD library kit on 86 F_2_ populations and their parents and sequenced on MGISEQ-2000RS using PE (150 bp) chemistry. A total of 261.9 Gbp reads were obtained (an average of 3.13 Gbp per sample); each read contained 150 bp (×2). Among these data, 85.57% of bases were high quality (>Q30), and the average guanine-cytosine (GC) content was 38.76%. The sequencing depth corresponded to 21.61-fold in TPT9, 14.71-fold in W054, and 25.62-fold in the 86 F_2_ progenies (Appendix A). A total of 23,576 polymorphic SNP markers were identified from the two parents. There were four types of polymorphic SNP markers: aa×bb, hk×hk, lm×ll, and nn×np. Since both of the parents were highly inbred, only the aa×bb genotypes showing homozygous and polymorphic between the parents were used in further analysis.

#### 2.3.2. Construction of the Winged Bean Genetic Linkage Map

While a total of 193,735 SNPs were identified, only the aa×bb marker could be used in the linkage analysis. After SNPs with a minor allele frequency >0.1, depth coverage between 10×–200×, fewer than 10% missing data, non−segregation distortion, and non−redundant genotypes were filtered, 1384 SNP markers remained and were constructed into a genetic linkage map. The map of the F_2_ population contained nine linkage groups (LGs) (Appendix A) and spanned a cumulative distance of 4552.8 cM (Table 2). The length of each LG ranged from 164.96 cM (LG6) to 755.74 cM (LG2), with an average of 505.87 cM. The number of SNP markers mapped in each linkage group varied from 36 markers on LG06 to 218 markers on LG02, with an average of 153.78 SNPs per LG (Table 2). The average distance between adjacent markers across the nine LGs was 3.38 cM (Table 2).

### 2.4. QTL Analysis for Flower-, Pod-, and Seed-Related Traits 

Two methods, ICIM implemented in software QTL IciMapping 4.1 and MQM implemented in MapQTL 5.0, were used for QTL analysis. For ICIM for which a LOD score of 3.0 was set as significant threshold for QTL analysis, a total of 31 significant QTLs on seven LGs were identified for the nine traits related to flower, pod, and seed. No QTL was detected on LG6 or LG9 (Table 3 and Figure 3). The additive effect for almost all QTLs was positive, indicating that the allele(s) from W054 increase values of these traits (Table 3). In the case of MQM, for which a significant LOD threshold was computed by a 1000-permutation test, only 8 significant QTLs were detected and no QTL was found for seed length and seed width (Table 4 and Figure 4). The QTLs were on LG1, LG2 and LG3, and most of them are on the LG1. The major QTL of each trait was always detected by both methods. Nonetheless, details of the QTLs detected by ICIM are described.

#### 2.4.1. QTLs Controlling Organ Coloration

Two QTLs, *qAntho1.1* and *qAntho5.1*, locating on LGs 1 and 5, were identified for pod anthocyanin content. These QTLs accounted for 28.92% and 11.61% of the anthocyanin content variation in the F_2_ population, respectively. At the *qAntho1.1* the allele(s) from W054 increased anthocyanin content, whereas at the *qAntho5.1* the allele(s) from TPT9 decreased the anthocyanin content. 

Three QTLs located on LGs 1 and 7 were detected for pod color. They explained 9.67% to 49.15% of pod color intensity variation in the F_2_ population. The QTL on LG1, *qPdc1.1*, showed the largest effect was on the LG1. At all QTLs, allele(s) from W054 increased coloration of the pod. The *qPdc1.2* expressed an overdominant gene effect. 

Similar to the pod color, three QTLs localizing on LGs1 and 7 were identified for calyx color. These QTLs explained 12.67% to 38.28% of calyx color variation in the F_2_ population. The QTL *qClxc1.1* located on LG1 showed the largest effect towards the calyx color. At these loci, allele(s) from W054 increased coloration of the pod. The *qClxc1.2* showed a very strong overdominant gene effect. 

Five QTLs residing on LGs 1, 3, and 7 were detected for wing color. They accounted for 7.76% to 18.61% of the wing color variation in the in the F_2_ population. Among these QTLs, *qWingc3.1* residing on LG3 showed the largest effect. However, the *qWingc7.1* mapped on LG7 showed comparable effect with the *qWingc3.1*. Allele(s) from W054 at the *qWingc1.1* and *qWingc7.2* increased coloration of the wing. In contrast, allele(s) from W054 at the *qWingc1.2*, *qWingc3.1* and *qWingc7.1* reduced coloration of the wing. The *qWingc1.2* showed a very strong overdominant gene effect. 

Seven QTLs located on LGs 1, 3, 4, 5 and 7 were identified for banner color. They explained 5.53% to 17.75% of the banner color variation in the in the F_2_ population. The QTL on LG3, *qBannerc3.1*, showed the greatest effect towards the banner color, followed by the *qBannerc3.2* that was mapped only 5.0 cM away from the *qBannerc3.1*. Allele(s) from W054 at the W054 at the *qBannerc1.2*, *qBannerc1.2*, *qBannerc3.2* and *qBannerc7.1* enhanced coloration of the banner. Allele(s) from W054 at the *qBannerc3.1*, *qBannerc4.1*, and *qBannerc5.1* reduced coloration of the banner. The *qBannerc3.2*, *qBannerc4.1* and *qBannerc5.1* expressed a strong overdominant gene effect, especially the *qBannerc3.2*. 

One major and one minor QTL were identified for seed coat color. The major QTL, *qSdc1.1*, was localized on LG1, while the minor QTL, *qSdc3.1*, was located on LG3. They explained 73.34% and 5.45% of the seed coat variation in the F_2_ population, respectively. Allele(s) from W054 at the *qSdc1.1* increased intensity of the seed coat color, whereas allele(s) from W054 at the *qSdc3.1* decreased the color intensity. The *qSdc1.1* expressed an overdominant gene effect. 

#### 2.4.2. QTLs Controlling Organ Size

Four QTLs for pod length were detected on LGs 1, 2, and 8. These QTLs accounted for 9.449% to 16.17% of the pod length variation in the F_2_ population. Among the four QTLs, *qPdl2.1* showed the greatest effect. At all these QTLs, allele(s) from W054 enhanced the length. However, all of the QTLs except *qPdl8.1* expressed overdominant gene effect.

Only one QTL was detected for seed length, *qSdl7.1*. This QTL was located on LG7 and explained 18.83% of the variation of seed length in the F_2_ population. Four QTLs on LGs 1 and 2 were detected for seed width, of which three QTLs were on the LG1. The QTLs explained 11.47% to 18.52% of the variation of seed width in the F_2_ population. The *qSdw2.1* on LG02 showed the highest effect. In contrast to other QTLs, at the *qSdw2.1* allele(s) from W054 decreased width of the seed. In addition, the *qSdw2.1* expressed a very strong overdominant gene effect. 

## 3. Discussion

### 3.1. Genetics Controlling Traits in Winged Bean 

There are only a few reports on inheritance of traits in winged bean. The winged bean parents, W054 and TPT9, used in this study were strikingly different in several traits related to flower, pod, and seed. Pod anthocyanin content, pod length, seed width, and seed length were considered as quantitative traits, while pod color, calyx color, wing color, banner color, and seed color were considered qualitative traits. Interestingly, these qualitative showed high variation in the F_2_ mapping population (Figure 2). Some F_2_ individuals showed color variants that did not present in their parents, particularly flower color (wings and banner) (Figure 2D,E). Transgressive segregation of these traits suggested that coloration in those organs is controlled by more than one gene and that W054 and TPT9 each possess different genetic loci contributing to coloration in those organs. A previous study demonstrated that pod wing color is controlled by a single gene in which purple is dominant to green [29]. In contrast, our study showed that green pod wing is dominant to purple pod wing (Figure 5A). Unfortunately, we did not investigate pod wing color in our F_2_ population. Additional study is necessary to clarify the contrasted findings between the two studies.

In this study, correlation analysis of the flower-, pod-, and seed-related traits in the F_2_ population was determined (Table 1). Moderate to high and significant correlations were found between coloration of these traits (Table 1). These results suggest that these traits are controlled by pleiotropic locus/loci or linked loci. It has been reported that genes controlling stem color and calyx color, and pod wing color and pod speck color are linked [30]. 

### 3.2. The Winged Bean Genetic Linkage Map

A genetic linkage map is indispensable for QTL analysis and gene discovery for marker-assisted breeding. The SNP-based linkage map constructed for the F_2_ population in this study is the first genetic map of the winged bean. The map comprised 1384 SNP markers that were assigned to 9 linkage groups corresponding to haploid chromosomes of the winged bean. The total length of the map was 4552.8 cM with average distance between markers of 3.38 cM which is sufficient for genome-wide QTL analysis for economically important traits [31]. In this study, RAD-seq was used to discover SNP markers. RAD-seq has been used for large-scale SNP discovery and genetic mapping in many crop species, especially in species without a reference genome [32,33,34]. The high-density linkage map constructed for winged bean with the SNPs discovered from RAD-seq indicated that this genotyping method can be applied to develop large-scale SNP markers, and construct high-density molecular genetic maps for winged beans without availability of reference genome. The method greatly shortens the sequencing cycle, and reduces the cost of marker development for non-model species [35]. 

### 3.3. QTL Analysis for Flower-, Pod-, and Seed-Related Traits of the Winged Bean 

In this study, QTL analysis using two methods, ICIM and MQM, in the F_2_ population revealed 31 QTLs in total for nine flower-, pod-, and seed-related traits (Table 3 and Table 4). The number per QTL ranged from one (seed length) to seven (banner color). Most of the traits investigated for the QTLs were associated with coloration of organs. The QTLs for organ colorations were clustered on LGs 1, 3 and 7 in which some QTLs were mapped to the same or nearly the same position (Table 3 and Table 4): for example, *qAntho1.1* for pod anthocyanin contents, *qPdc1.1* for pod color and *qClxc1.1* for calyx color (Table 3) and *qWingc1.1* for wing color and *qSdc1.1* for seed color (Table 4). However, on each LG, different QTLs were also found for the same trait (Table 3). These results suggested that organ coloration in the winged bean is controlled by several QTLs in which some of the traits are likely to be controlled by the same gene(s) (pleiotropic locus) and/or by the linked genes. This is in line with the significant and positive correlation among organ coloration traits in the F_2_ population (Table 1). The results also suggested that anthocyanin is the major cause of purple pod color and purple calyx color in winged bean. 

Among organ coloration traits in the winged bean, the color of young pods is associated with consumer and farmer preferences [10]. In Thailand, most of winged bean cultivars produce green pods, while cultivars with purple pods are rare, especially the ones with dark purple pods [10]. Purple coloration in plants is principally caused by anthocyanin pigments which are associated with health benefits to humans [36,37]. Recently, winged cultivars with purple coloration have become more popular among consumers because of health effects of anthocyanins. In this study, QTL analysis showed that anthocyanin contents in the young pods is controlled by only two major QTLs, *qAntho1.1* and *qAntho5.1* (Table 3). This suggests that breeding new winged bean cultivars with purple young pods may not be difficult. However, interestingly, additive effects towards pod color of the W054 (the parent with dark purple pods) at the *qAntho1.1* and *qAntho5.1* showed different directions; increasing anthocyanin content at *qAntho1.1*, while decreasing anthocyanin content at *qAntho5.1*. This may complicate the selection for dark purple pods. The parents, W054 and TPT9, were different in flower color: purple vs. light blue. However, the F_2_ populations showed wide variants in flower colors (Figure 5C). As many as 12 QTLs were detected for flower color, 5 for wing color and 7 for banner color. Two co-localizations of QTLs were found for wing and banner colors; one on LG1 (*qWingc1.1* and *qBannerc1.1*) and one on LG7 (*qWingc7.1* and *qBannerc7.1*) (Table 3). Both purple and blue colors in plants are due to presence of anthocyanin pigments [36,37]. Cyanidin, delphinidin, pelargonidin, peonidin, malvidin, and petunidin are the most common anthocyanidins distributed in the plants [36]. While most of these anthocyanidins give rise to purple or red colors in plants and foods, malvidin and delphinidin provide blue-colored flowers [38,39]. The QTLs identified for flower color in the winged bean may be associated with production of different anthocyanidins. 

Apart from pod color, pod length is also associated with consumer’s and farmer’s preferences [10]. Although long and extralong pods are possibly advantageous for pod weight yield and some local consumers and farmers like winged bean cultivars with long and extralong pods, cultivars with short pods are much more popular because of convenience in packaging. In fact, cultivars having extralong pods are scarce [10]. In this study, pod length of the mapping parents W054 and TPT9 were about 10 cm in difference and four QTLs were identified for this trait. Interestingly, the major QTL *qPdl1.1* controlling pod length was mapped to nearly the same position with the major QTL *qAntho1.1* conferring pod anthocyanin content (Table 3). This suggests that increasing pod length would not hamper purple pod coloration. Nonetheless, fine mapping is necessary to elucidate the co-localization of the *qPdl1.1* and *qAntho1.1.* It is note-worthy that the *qPdl2.1*, the largest effect QTL for pod length, expressed a strong over-dominance gene effect (potence ratio is about 4.1; Table 3). Although this strong overdominance effect may be useful for deployment of hybrid cultivars of winged bean, it can delay development of pure line cultivars in the winged bean as this crop is self-pollinating. Nonetheless, since overdominant effect can stem from allelic interaction or closely-linked genes [40], additional study is necessary to determine the cause of strong overdominance of the *qPdl1.1*. Moreover, a large population of recombinant inbred lines should be used for confirmation or fine mapping of the QTLs detected in this study.

## 4. Materials and Methods

### 4.1. Plant Materials 

An F_2_ mapping population was developed from a cross between TPT9 and W054. TPT9, an accession originally from Nigeria, has a short green pod, whereas W054 is a commercial winged bean cultivar from the Nakhon Pathom province, Thailand, containing a long purple pod (Figure 5). TPT9 is used as the male parent, crossed onto W054 to develop F_1_ hybrids. An F_1_ hybrid was self-pollinated to produce F_2_ seeds. Eighty-six F_2_ individuals were grown under a field condition for DNA extraction and phenotypic evaluation. 

### 4.2. Field Evaluation

The F_2_ mapping population and the parents were sown in a row with 1 m intra-hole spacing and 1 m inter-row spacing. The row consisted of 50 plants (W054 and TPT9) which were subjected to a randomized complete block design (RCBD) with non-replication. The experimental field was located at the Agronomy Field Crops Station, Khon Kaen University, Khon Kaen, Thailand, and used from August 2020 to April 2021. 

Measurement of pod length: pod length was determined in the F_2_ individuals. Three to five fresh pods were harvested from individual F_2_ plant at 15 days after flowering (DAF) for pod length (cm) measurement.

Flower-related traits: three flowers of the F_2_ individuals were observed for banner, wing, and calyx colors using color scoring following IPGRI [41] with a slight modification as white, white blue, white purple, blue, violet, and purple (Figure 5C). 

Seed-related traits: ten seeds of the F_2_ individuals were observed for color using color scoring following IPGRI [41]. Seed length and seed width (cm) were also recorded. 

Measurement of pod pigmentation: pigmentation of the F_2_ pods was recorded using color scoring following IPGRI [41] with a slight modification as green, green with purple spots, green with a purple spot at center, purple wing, purple center, light purple, purple, and dark purple (Figure 5B), and the pod anthocyanin contents were quantified. The sample preparation and extraction of anthocyanin followed a slightly modified method as previously reported [42]. The pH differential method was used to measure total anthocyanin content (TAC) [43]. Briefly, three fresh pods of each individual F_2_ plant were sliced (0.2 g) and submerged in 10 mL of 0.1% trifluoroacetic acid (TFA) and 95% ethanol in a 50 mL tube for anthocyanin extraction. The components were incubated at 4 °C for six hours until the solution was colorless. The extract was filtrated through qualitative cellulose filter paper (Whatman No.1, Sigma-Aldrich^®^, St. Louis, MO, USA). The anthocyanin of the filtrate was evaluated by UV-Vis spectrophotometer at 530 nm [42]. The total anthocyanin content was calculated by the following formula: (1)$$\mathrm{TAcy}=\mathrm{OD}\;\times \;\mathrm{DV}\frac{100\times \;\mathrm{TEV}}{\mathrm{SV}\;\times \;\mathrm{SW}\;\times \left(\frac{\mathrm{Ecf}}{10}\right)}$$
where,
TAcy = total anthocyanin (mg/100 g)OD = Absorbance 530 nmDV = Volume of diluted solution (mL) SV = Volume of extracted solution for diluted (mL)TEV = Total extracted volume (mL)SW = Sample weight (g)Ecf = Extinction coefficient (Ecf of cyaniding-3-glucoside is 449.2)

### 4.3. RAD-Seq

Total genomic DNA of the parents and F_2_ population was extracted from fresh leaf tissue using the CTAB method described by Lodhi et al. [44]. The DNA quantity and quality were determined by gel electrophoresis and spectrophotometric measurements. Total genomic DNA (300 ng) was used to construct a RAD-seq [28] library following the protocol in the MGIEasy RAD library preparation kit (MGI Tech). Paired-end (150 bp) sequencing was performed on the MGISEQ-2000RS (MGI Tech) according to the manufacturer’s instructions. SNPs were called using Stacks ver. 2.58 [45] and filtered with the following criteria: (a) a minor allele frequency >0.1, (b) depth coverage between 10×–200×, and (c) fewer than 10% missing data. In addition, markers showing significant segregation distortion (χ^2^ test *p*-value < 0.01) and having same genotypes as large haplotype blocks were excluded from further analysis. The filtered SNP markers were used to construct linkage maps. 

### 4.4. Linkage Map Construction

A linkage map was constructed utilizing QTL IciMapping 4.2 [46]. The segregation ratio of each SNP marker was determined by chi-square test. Markers showing significant (*p* < 0.05) segregation distortion were excluded from linkage analysis. Then, markers were grouped using log of odds (LOD) value of 7.0. Markers on the same linkage were ordered by recombination counting and ordering (RECORD) algorithm [47] with 2-optMAP function [48]. Map distances were calculated using the Kosambi mapping function [49].

### 4.5. QTL Analysis

QTLs for flower-, pod-, and seed-related traits were identified by inclusive composite interval mapping (ICIM) method [50] implemented in QTL IciMapping 4.1 software [46]. ICIM is performed at every 0.1 cM with a probability in stepwise regression (PIN) of 0.001. LOD threshold of 3.0 was used to declare the presence of QTLs for each trait. In addition, the QTLs for all the traits were also identified by multi-QTL model (MQM) method using MapQTL 5.0 software [51]. A 1000 permutation test was used to determine the LOD score significance thresholds at a 95% confidence level.

## 5. Conclusions

In this study, we constructed the first genetic linkage map of the winged bean using SNP markers generated from next generation sequencing technology. We performed QTL mapping for flower color, pod color and length, and seed color and seed size for this crop. The mapping developed in this study is moderately saturated with distance between markers of about 3.0 cM and resolved haploid chromosome number of the winged bean. Several QTLs related to the traits mentioned above were mapped on to the linkage map of which some QTLs were co-localized. This linkage map is useful for molecular breeding studies of the winged bean, while QTLs identified in this study provide better understanding of the genetics of organ coloration in the winged bean.

## Figures and Tables

**Figure 1 plants-11-00500-f001:**
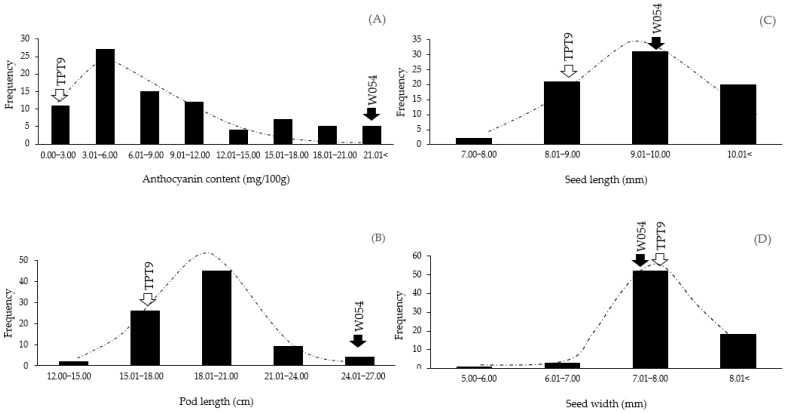
Frequency distribution of pod anthocyanin content (**A**), pod length (**B**), seed length (**C**), and seed width (**D**) of the winged bean F_2_ population derived from the cross W054 × TPT9.

**Figure 2 plants-11-00500-f002:**
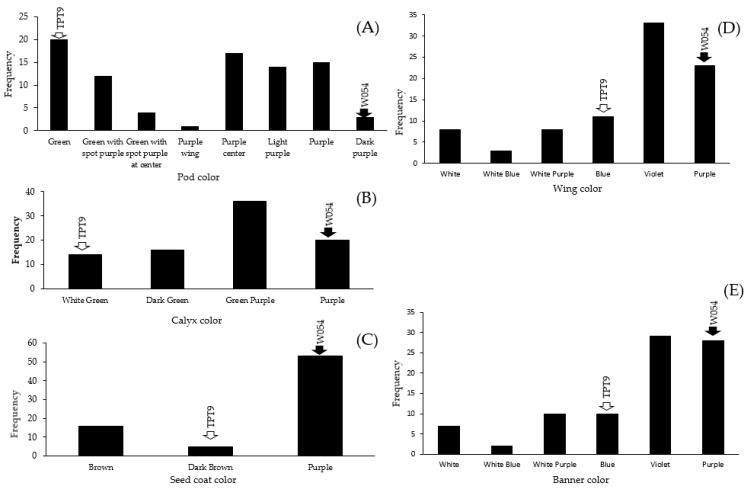
Frequency distribution of coloration of pod (**A**), calyx (**B**), seed (**C**), wing (**D**), and banner (**E**) in the winged bean F_2_ population derived from the cross W054 × TPT9.

**Figure 3 plants-11-00500-f003:**
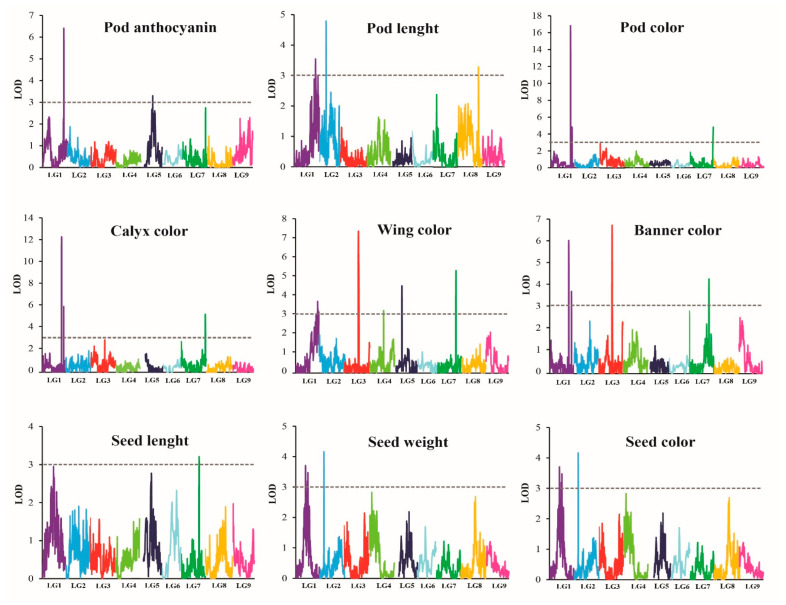
LOD (Logarithm of odds) graphs of quantitative trait loci for flower-, pod-, and seed-related traits detected by inclusive composite interval mapping (ICIM) method in the winged bean F_2_ population developed from the cross W054 × TPT9. The *x*-axis indicates the linkage groups, whereas the *y*-axis indicates the logarithm of odds (LOD) scores. The dash line horizontal to the *y*-axis indicates the LOD significance threshold.

**Figure 4 plants-11-00500-f004:**
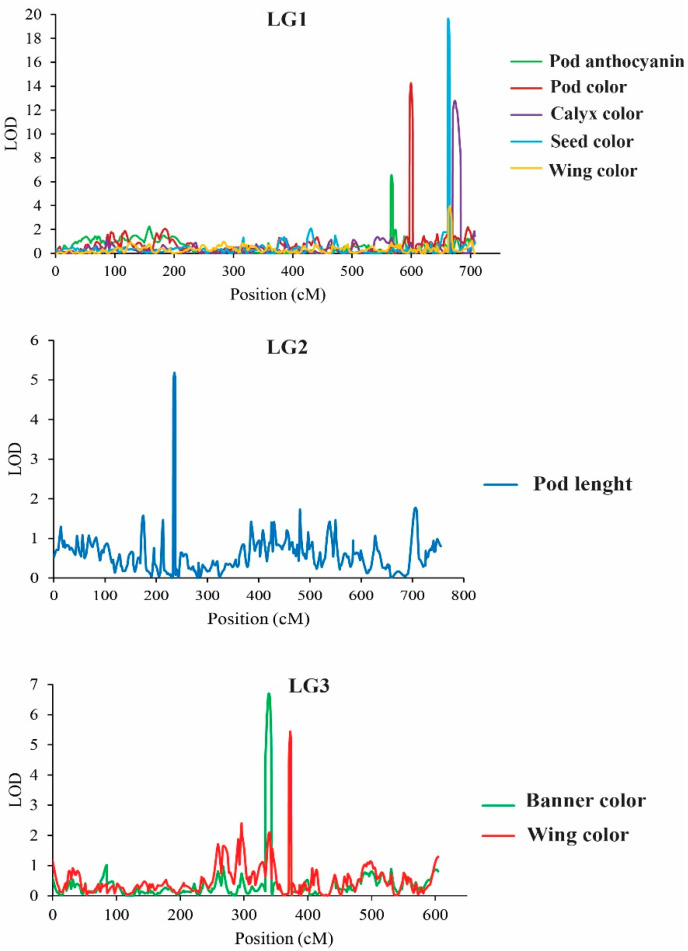
LOD (Logarithm of odds) graphs of quantitative trait loci for flower-, pod-, and seed-related traits detected on LG1, LG2 and LG3 by multiple QTL mapping (MQM) method in the winged bean F_2_ population developed from the cross W054 × TPT9. The *x*-axis indicates the linkage groups, whereas the *y*-axis indicates the logarithm of odds (LOD) scores. The dash line horizontal to the *y*-axis indicates the LOD significance threshold.

**Figure 5 plants-11-00500-f005:**
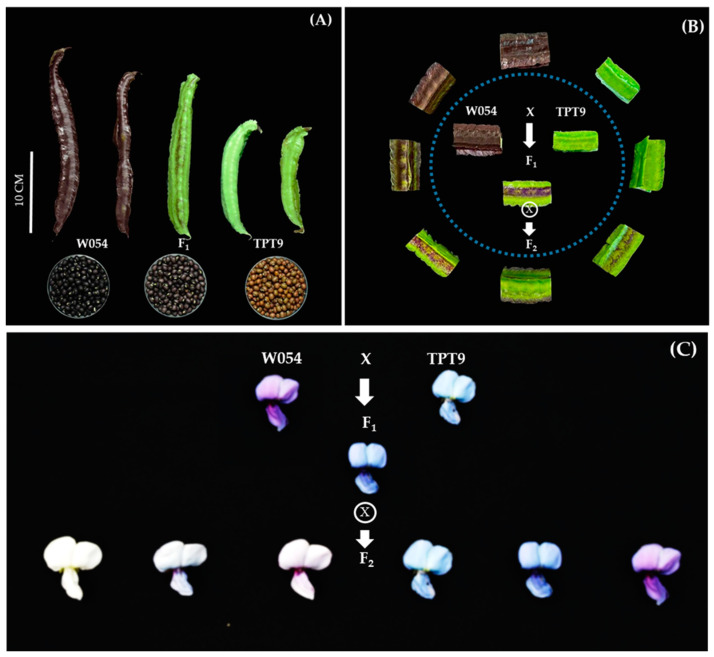
The phenotypes, parental and progenies, of the W054 × TPT9 cross. The pods and seeds of W054, TPT9, and F_1_ progeny are shown in (**A**), the pod colors of W054, TPT9, F_1_ progeny, and diversity of eight types are located in the surrounding F_2_ population (**B**), and the wing and banner colors of W054, TPT9, F_1_ progeny, and six types of F_2_ population are shown in (**C**).

**Table 1 plants-11-00500-t001:** Correlation between pod anthocyanin content, pod length, seed length, seed width; and pod, seed, calyx, wing, and banner colors of the winged bean F_2_ population derived from cross W054 × TPT9.

Traits	AnthoCy	PDL	PDC	CalyxC	WingC	BannerC	SeedL	SeedW
PDL	0.248 *							
PDC	0.748 **	0.329 **						
CalyxC	0.706 **	0.262 *	0.812 **					
WingC	0.431 **	0.091	0.421 **	0.543 **				
BannerC	0.419 **	0.081	0.405 **	0.523 **	0.978 **			
SeedL	0.158	0.151	0.110	0.095	0.082	0.114		
SeedW	−0.034	−0.008	0.010	−0.027	−0.110	−0.096	0.500 **	
SeedC	0.381 **	0.239 *	0.547 **	0.595 **	0.400 **	0.378 **	−0.018	−0.197

AnthoCy = pod anthocyanin content, PDL = pod length, SDL = seed length, SDW = seed width, PDC = pod color, SDC = seed coat color, CalyxC = calyx color, WingC = wing color, and BannerC = banner color. * Significant at 0.05 probability. ** Significant at 0.01 probability.

**Table 2 plants-11-00500-t002:** Characteristics of an SNP-based linkage map constructed for winged bean F_2_ population of 86 individuals derived from the cross W054 × TPT9.

Linkage Group	Number of SNP Markers	Length (cM)	Average Marker Interval (cM)	Maximum Interval (cM)
1	208	707.12	3.40	11.09
2	218	755.74	3.47	11.98
3	186	604.61	3.25	13.37
4	170	507.73	2.98	11.53
5	96	293.99	3.06	9.55
6	36	164.96	4.58	16.44
7	176	560.28	3.18	10.67
8	180	587.78	3.27	16.90
9	114	370.59	3.25	15.28
Average	153.78	505.87	3.38	-
Total	1384	4552.8	-	-

**Table 3 plants-11-00500-t003:** Details of the QTLs detected for pod flower-, pod- and seed-related traits in an F_2_ population of winged bean derived from the cross W054 × TPT9 by inclusive composite interval mapping method.

Traits	QTLs	LGs	Positions	Marker Interval	LOD Score	PVE (%)	Add	Dom
Anthocyanin content	*qAntho1.1*	1	599.2	90362_173–86339_105	6.41	28.92	4.41	1.47
	*qAntho5.1*	5	137.4	129210_17–74111_70	3.29	11.61	−2.16	−2.81
Pod length	*qPpl1.1*	1	601.3	86339_105–88581_215	3.54	11.08	0.86	1.18
	*qPdl1.1*	1	609.8	88581_7–113679_228	3.03	9.49	0.88	1.04
	*qPdl2.1*	2	236.6	65543_37–65543_66	4.79	16.17	0.44	1.79
	*qPdl8.1*	8	503.0	758788_184–45825_50	3.39	11.22	1.39	0.01
Pod color	*qPdc1.1*	1	599.4	86339_105–88581_215	16.82	49.15	2.22	0.15
	*qPdc1.2*	1	643.9	50448_148–73516_244	4.84	10.35	0.06	1.56
	*qPdc7.1*	7	560.2	6763_192–141967_88	4.81	9.69	1.02	−0.47
Calyx color	*qClxc1.1*	1	599.4	86339_105–88581_215	12.26	38.28	0.76	0.05
	*qClxc1.2*	1	664.9	5563_75–37621_162	5.84	15.01	0.08	0.74
	*qClxc7.1*	7	543.1	1977_174–6763_192	5.12	12.67	0.44	−0.23
Wing color	*qWingc1.1*	1	569.0	49233_195–144410_63	6.01	16.33	0.76	0.03
	*qWingc1.2*	1	649.8	73516_104–64897_31	3.67	9.72	−0.05	0.85
	*qWingc3.1*	3	340.7	59095_37–59095_183	6.72	18.61	−0.93	0.05
	*qWingc7.1*	7	0.0	92184_83–54946_220	3.17	7.77	−0.48	0.29
	*qWingc7.1*	7	435.3	79824_126–57556_229	4.85	12.77	0.79	−0.02
Banner color	*qBannerc1.1*	1	670.1	37621_240–101902_163	3.65	6.45	0.32	0.66
	*qBannerc1.2*	1	706.5	71361_136–71361_18	3.09	5.54	0.48	0.52
	*qBannerc3.1*	3	340.1	120628_171–59095_37	7.34	17.75	−1.02	0.10
	*qBannerc3.2*	3	345.1	59095_229–542971_120	6.12	14.84	0.02	1.21
	*qBannerc4.1*	4	283.3	29428_113–114955_133	3.17	6.66	−0.18	0.79
	*qBannerc5.1*	5	81.4	42289_151–31093_166	4.46	9.57	−0.24	0.99
	*qBannerc7.1*	7	435.3	79824_126–57556_229	5.27	11.51	0.85	0.32
Seed length	*qSdl7.1*	7	414.7	143811_217–2570_116	3.21	18.83	0.48	0.12
Seed width	*qSdw1* *.1*	1	294.8	79225_198–130456_96	3.70	14.43	0.27	0.29
	*qSdw1* *.2*	1	347.9	88187_3–100448_14	3.19	11.47	0.23	0.29
	*qSdw1.3*	1	367.2	46319_242–77017_78	3.47	12.41	0.20	0.30
	*qSdw2.1*	2	134.6	10683_187–3208_238	4.16	18.52	−0.09	0.51
Seed color	*qSdc1.1*	1	660.9	64897_73–73327_179	22.74	73.33	0.78	0.89
	*qSdc3.1*	3	277.3	74742_203–41532_216	3.27	5.45	−0.30	0.01

LG = linkage group; PVE = phenotypic variance explained by the QTL; Add = additive effect; Dom = dominant effect.

**Table 4 plants-11-00500-t004:** Details of the QTLs detected for pod flower-, pod- and seed-related traits in an F_2_ population of winged bean derived from the cross W054 × TPT9 by multiple QTL mapping method.

Traits	QTLs	LGs	Positions	Marker Interval	LOD Score	PVE (%)	Add	Dom
Anthocyanin	*qAntho1.1*	1	566.5	59413_160–49233_195	6.57	29.60	4.57	2.15
Pod length	*qPpl2.1*	2	235.5	69533_212–65543_37	5.18	24.20	0.51	2.03
Pod color	*qPdc1.1*	1	599.4	86339_105–88581_215	14.26	53.40	2.15	1.39
Calyx color	*qClxc1.1*	1	673.03	37621_240–101902_163	12.78	49.60	0.82	0.83
Wing color	*qWingc1.1*	1	664.83	73327_66–5563_75	3.94	19.00	0.72	0.68
	*qWingc3.1*	3	372.46	52207_209–15865_163	5.44	25.30	−0.76	0.96
Banner color	*qBannerc3.1*	3	338.69	120628_171–59095_37	6.69	30.10	−0.94	1.14
Seed length	No QTL detected							
Seed width	No QTL detected							
Seed color	*qSdc1.1*	1	662.34	73327_179–73327_66	19.65	70.60	0.78	0.87

LG = linkage group; PVE = phenotypic variance explained by the QTL; Add = additive effect; Dom = dominant effect.

## Data Availability

Data presented in this study are available on request from the corresponding author.

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
