# Peer review of "The First Genetic Linkage Map of Winged Bean [Psophocarpus tetragonolobus (L.) DC.] and QTL Mapping for Flower-, Pod-, and Seed-Related Traits"

_plants, 2022, doi:10.3390/plants11040500_

Round 1

Reviewer 1 Report

The manuscript is well written and the results are sound and well presented. There are some minor issues to be addressed by the authors:

Lines 113-114 and 129-130. How the authors reached to the conclusion that in the first case the traits examined are following a normal distribution and are controlled by polygenes, while in the second case are showing a different mode of inheritance? Did they performed any statistical test to examine the distribution of those traits?

Line 156: how they explain the low sequencing depth in W054. Was it a technical issue or it due to the DNA base composition? Did this low sequence depth affect the analysis when compared to the other samples?

Lines 158-159: The genotypes need to be better presented and explained.

Lines 163-164: They need to rephrase and explain beer what they did and how they filtered out the SNPs.

There is a discrepancy between lines 418 Funding and 421 Ackgnowledgments

Author Response

Comments

Responses

The manuscript is well written and the results are sound and well presented. There are some minor issues to be addressed by the authors:

Thank you for your comment. We revised the manuscript followed your suggestions.

Lines 113-114 and 129-130. How the authors reached to the conclusion that in the first case the traits examined are following a normal distribution and are controlled by polygenes, while in the second case are showing a different mode of inheritance? Did they performed any statistical test to examine the distribution of those traits?

The distribution of quantitative traits such as pod anthocyanin content, pod length, seed length, and seed width in the F2 population were normally distributed (Figure 1) (Durbin-Watson Statistic test with P-values = 0.7053, 0.2435, 0.9836, and 0.1926, respectively) (Lines 114-116). Generally, the continuous and normal distribution suggest that these traits are controlled by multiple loci.

Line 156: how they explain the low sequencing depth in W054. Was it a technical issue or it due to the DNA base composition? Did this low sequence depth affect the analysis when compared to the other samples?

Thank you for your comment.

We used RADseq technique which reduced representative portion of the genome by 100x and only sequenced around 1% of the genome.  From our experience, for an organism with a genome size of around 1 Gb, the number of called SNP from RAD-seq reads will start to saturate from 5 million reads.  In this case, both parents have >10 million reads; therefore, we are certain that the quality and quantity of SNP would be sufficient. Furthermore, since winged bean has no reference genome, we used the Stacks program to de novo assemble raw reads into contigs which were then selected and used as reference contigs for mapping and calculating coverage.  The coverage calculation is weighted by the number of loci in each sample as well.

Lines 158-159: The genotypes need to be better presented and explained.

Th sentence in the lines 158-159 is correct.  It is the way the program (JoinMap) calls different types of genotype to avoid using the same letters from different types of genotype. We explained for the information of genotyping as” There were four types of polymorphic SNP markers: aaxbb, hkxhk, lmxll, and nnxnp. Since both of the parents were highly inbred, only the aaxbb genotypes showing homozygous and polymorphic between the parents were used in further analysis”.

Lines 163-164: They need to rephrase and explain beer what they did and how they filtered out the SNPs.

Thank you for your comment. We revised as “After SNPs with a minor allele frequency >0.1, depth coverage between 10x -200x, fewer than 10 % missing data, non-segregation distortion, and non-redundant genotype were filtered” [Lines 170-172]. And also in M&M as “SNPs were called using Stacks ver. 2.58 [46] and filtered with the following criteria: a) a minor allele frequency >0.1; b) depth coverage between 10x -200x; and c) fewer than 10 % missing data. In addition, markers showing significant segregation distortion (χ2 test p-value < 0.01) and having same genotypes as large haplotype blocks were excluded from further analysis”. (Lines 438-442)

There is a discrepancy between lines 418 Funding and 421 Acknowledgments

Thanks, we revised for consistency of funding and acknowledgments (Lines 418 and 421)

Reviewer 2 Report

Chankaew et al. (plants-1577981) used the RAD-seq method to identify the SNP without reference genome sequencing of winged bean. They conducted the QTL mapping with ICIM analysis for flower-, pod-, and seed-related traits. QTL analysis in the F2 population revealed 31 QTLs controlling pod length, pod color, pod anthocyanin content, flower color, and seed color.

To improve this manuscript, I have several comments. 

Line 58. “number of landrace winged bean cultivars” means cultivar or landrace? It makes me confused.

Line 158. Does “aaxbb, hkxhk, lmxll and nnxnp” mean aaxbb, abxab, abxbb, and aaxab? If not, please mention it clearly.

Line 162. As you know, your population is small. I think the missing percentage is too low to filter SNPs out. And 1,384 SNP was your final number of SNPs from 193,735 SNPs. 1 % of total called SNPs were used in this study. If you selected more SNPs for QTL mapping, I guess your result can be improved. As you have used the F2, you can not have replication. Better to use RIL or F2:3 population. Another issue is the population size is small. Please consider it for the next study. 

Figure 3 should move the supplementary figure. A new figure is required. I would like you to make a genetic map with QTL regions using MapChart software. Please see figure 3 of https://doi.org/10.1007/s00122-010-1385-7

Please use MapQTL software or QTL cartographer to confirm the QTL regions from ICIMapping software. Make a figure for the results from MapQTL software or QTL cartographer.

Author Response

Comments

Responses

Line 58. “number of landrace winged bean cultivars” means cultivar or landrace? It makes me confused.

Thank you. We revised it.

Line 158. Does “aaxbb, hkxhk, lmxll and nnxnp” mean aaxbb, abxab, abxbb, and aaxab? If not, please mention it clearly.

That is correct.  It is the way the program (JoinMap) calls different types of genotype to avoid using the same letters from different types of genotype.

We explained for the information of genotyping as” Polymorphic SNP marker contained four genotypes: aaxbb, hkxhk, lmxll, and nnxnp.  Since both of the parents were highly inbred, only the aaxbb genotypes showing homozygous and polymorphic between the parents were used in further analysis. (Lines 162-165)

Line 162. As you know, your population is small. I think the missing percentage is too low to filter SNPs out. And 1,384 SNP was your final number of SNPs from 193,735 SNPs. 1 % of total called SNPs were used in this study. If you selected more SNPs for QTL mapping, I guess your result can be improved. As you have used the F2, you can not have replication. Better to use RIL or F2:3 population. Another issue is the population size is small. Please consider it for the next study.

Thanks for your comment. We revisesd in M&M as “SNPs were called using Stacks ver. 2.58 [46] and filtered with the following criteria: a) a minor allele frequency >0.1; b) depth coverage between 10x -200x; and c) fewer than 10 % missing data. In addition, markers showing significant segregation distortion (χ2 test p-value < 0.01) and having same genotypes as large haplotype blocks were excluded from further analysis”. (Lines 438-442). In the next study, we will used a larger size of population for mapping.

Figure 3 should move the supplementary figure. A new figure is required. I would like you to make a genetic map with QTL regions using MapChart software. Please see figure 3 of https://doi.org/10.1007/s00122-010-1385-7

We moved the Figure 3 to supplemental file as per your suggestion. And we made a new figure showing a LOD graph of QTLs for all the traits (new Figure 3). However, we did not use MapChart because the software is unable to draw map with LOD graph of highly-dense map/markers. I hope this is acceptable for you.

Please use MapQTL software or QTL cartographer to confirm the QTL regions from ICIMapping software. Make a figure for the results from MapQTL software or QTL cartographer.

We followed your suggestion [Lines 456-58], please see Table 4 and Figure 4. However, we still used results from ICIM to describe the QTL results [Lines 181-191].

Round 2

Reviewer 2 Report

Thank you for your work.